# Comparison of Gold Nanoparticles Deposition Methods and Their Influence on Electrochemical and Adsorption Properties of Titanium Dioxide Nanotubes

**DOI:** 10.3390/ma13194269

**Published:** 2020-09-25

**Authors:** Ewa Paradowska, Katarzyna Arkusz, Dorota G. Pijanowska

**Affiliations:** 1Department of Biomedical Engineering, Faculty of Mechanical Engineering, University of Zielona Gora, Prof. Zygmunta Szafrana 4 Street, 65-516 Zielona Gora, Poland; k.arkusz@ibem.uz.zgora.pl; 2Nalecz Institute of Biocybernetics and Biomedical Engineering, Polish Academy of Sciences, Ks. Trojdena 4 Street, 02-109 Warszawa, Poland; dpijanowska@ibib.waw.pl

**Keywords:** titanium dioxide nanotubes, chronoamperometry, cyclic voltammetry, gold nanoparticles, electrochemical characteristic, bovine serum albumin adsorption

## Abstract

The increasing interest of attachment of gold nanoparticles (AuNPs) on titanium dioxide nanotubes (TNTs) has been devoted to obtaining tremendous properties suitable for biosensor applications. Achieving precise control of the attachment and shape of AuNPs by methods described in the literature are far from satisfactory. This work shows the comparison of physical adsorption (PA), cyclic voltammetry (CV) and chronoamperometry (CA) methods and the parameters of these methods on TNTs properties. The structural, chemical, phase and electrochemical characterizations of TNTs, Au/TNTs, AuNPs/TNTs are carried out using scanning electron microscopy (SEM), electrochemical impedance spectroscopy, X-ray diffraction, X-ray photoelectron spectroscopy. The use of PA methods does not allow the deposition of AuNPs on TNTs. CV allows easily obtaining spherical nanoparticles, for which the diameter increases from 20.3 ± 2.9 nm to 182.3 ± 51.7 nm as a concentration of tetrachloroauric acid solution increase from 0.1 mM to 10 mM. Increasing the AuNPs deposition time in the CA method increases the amount of gold, but the AuNPs diameter does not change (35.0 ± 5 nm). Importantly, the CA method also causes the dissolution of the nanotubes layer from 1000 ± 10.0 nm to 823 ± 15.3 nm. Modification of titanium dioxide nanotubes with gold nanoparticles improved the electron transfer and increased the corrosion resistance, as well as promoted the protein adsorption. Importantly, after the deposition of bovine serum albumin, an almost 5.5-fold (324%) increase in real impedance, compared to TNTs (59%) was observed. We found that the Au nanoparticles—especially those with smaller diameter—promoted the stability of bovine serum albumin binding to the TNTs platform. It confirms that the modification of TNTs with gold nanoparticles allows the development of the best platform for biosensing applications.

## 1. Introduction

The last decade has confirmed the preparation, characterization and application of various nanostructures such as nanotubes, nanoparticles, nanorods and nanowires—each with unique properties. Titanium dioxide (TiO_2_) nanotubes are materials used as a biosensor platform and possess several beneficial properties in sensor construction, e.g., the ease of preparation methods, high orientation, large surface area, high uniformity and excellent biocompatibility [1,2]. Increasing the support surface area directly impacts the effectiveness of biochemical reactions. Previous studies have been performed to obtain self-organized titanium dioxide nanotubes (TNTs) on titanium foil through electrochemical oxidation [3,4]. Compared to other modification methods, electrochemical anodization is considered the optimum approach because the nanotubes scale form can be easily be controlled by adjusting voltage, current, time, temperature and electrolyte components [5,6]. The electrical conductivity and adsorption properties depend on the morphology of TNTs and their modification process. According to our previous research [2,7], the best electrochemical properties show the TNTs, for which the diameter was equal to 50 nm. TNTs adsorption and electrical properties are improved by thermal processing and doping with gold nanoparticles. TiO_2_ content in amorphous formand crystalline form of rutile and/or anatase can be adjusted by annealing in high temperatures and different atmospheres [8,9,10]. The most desirable crystal phases are anatase and rutile from the three structures, which show good electrical conductivity [8]. Anatase crystallizes during annealing in the temperature ≥300 °C and the transformation into rutile occurs during thermal modification in the temperatures between 450 and 550 °C [8,9]. The modification of TNTs with gold nanoparticlesis very attractive. AuNPs attract researchers interest due to their good conductivity and biocompatibility [11,12]. The catalytic activity of AuNPs is a result of the quantum-scale dimension and the large surface-to-volume ratio [13].

Several methods were used to deposit AuNPs on TNTs. Direct adsorption [14] and soaking at high temperatures [15] are long-lasting processes (from 1 to 20 h), depending on the size and number of the deposited nanoparticles. Physical adsorption methods depend heavily on time and concentrations of solution that influence on reproducibility of the AuNP deposition process. The galvanostatic method described by Hosseini et al. [16,17] results in the formation of large-diameter nanoparticles. For example, for the same process parameters (current density of 10 mA/cm^2^, time: 5 min, temperature: 45 °C), the authors obtained 80–100 nm nanoparticles [16] and 100–200 nm [17]. Photoreduction is a long-lasting and difficult to precisely control parameters technique [18]. In turn, the use of cyclic voltammetry [19,20] and chronoamperometry [21] greatly simplify the process of modifying the TNTs platform. The authors’ earlier work showed the possibility of controlling the AuNPs morphologic parameters (diameter in the range of 20.3 ± 2.9 to 182.3 ± 51.7 nm) by choosing parameters of the CV process [22]. To date, there is no clear indication in the literature concerning the effect of time of the chronoamperometric method on the size and amount of AuNPs deposited on TiO_2_ nanotubes surface. There is also no information about the influence of this method on the TNT structure and comparison of gold-nanoparticle deposition methods. In order to further improve the adsorption properties and TNTs electrical conductivity, it is necessary to develop an electrode modified with spherical shaped gold nanoparticles with a high surface area-to-volume ratio.

AuNPs/TNTs arrays have many potential applications, so it is important to obtained well-configured AuNPs by direct adsorption, chronoamperometry and cyclic voltammetry methods. We focused on the structural, chemical, phase and electrochemical investigations of TNTs decorated with AuNPs. This research aimed to produce uniformly deposited, non-agglomerated and spherical AuNPs incorporated into TNTs, as well as TNTs covered by a so-called microlayer of gold used as a reference platform. Their impact on the morphology of TNTs and its conductivity was specified. In order to confirm the possibility of the use of AuNPs/TNTs to simplify the detection system without surface functionalization, the evaluation of adsorption of protein in the form of bovine serum albumin (BSA) was also verified. Albumin is a protein that makes up about 60% of all plasma proteins and has many biosensor and clinical applications. BSA has a structure similar to human serum albumin [23]. The AuNPs functionalization and adsorption of proteins like BSA improve the biodistribution and biocompatibility of AuNP nanostructures. BSA in the phosphate buffered saline (PBS) solution has a negative charge that allows its direct adsorption on a positive charged platforms like TNTs after AuNPs modification. BSA adsorption kinetics so far has not been studied. The research of BSA adsorption presented in this paper can also clearly indicate which of the techniques chronoamperometry (CA) or cyclic voltammetry (CV), diameter, and content of gold may result in obtaining the platform mostsensitive toBSA deposition, which is extremely important for biosensor applications. To estimate the selectivity of the AuNPs/TNTs relative to BSA-anti-BSA complex, interleukin-6 (IL-6), was used as a comparative protein. The effectiveness of BSA immobilization on TNTs and AuNPs/TNTs was determined based on microscopic and electrochemical characteristics.

## 2. Materials and Methods

### 2.1. Materials

Titanium foil with a chemical purity of 99.7%, ammonium fluoride with chemical purity ≥98%, phosphate buffered saline (0.01 M PBS, pH 7.4), ethylene glycol (assay 99.8%), gold (III) chloride hydrate HAuCl_4_∙3H_2_O (assay 99.995%), gold nanoparticles (10 nm stabilized suspension in 0.1 mM PBS), bovine serum albumin (BSA, purity ≥ 98%), were purchased from Sigma-Aldrich (St. Louis, MO, USA). All of the solutions were prepared from Milli-Q water.

### 2.2. Titanium Dioxide Nanotubes Preparation

Titanium foil—5 mm (width) × 15 mm (height) × 0.25 mm (thickness) was sonicated in acetone, next in distilled water and dried under liquid nitrogen atmosphere. TNTs were formed by anodizing of titanium foil [2]. The process of TNTs formed was conducted in ethylene glycol (85% wt.) with ammonium fluoride (0.65% wt.). The formation of titanium dioxide nanotubes was carried out at a potential of 17 V on an Autolab PGSTAT302N Metrohm (Herisau, Switzerland) for a 3750 s. In this case, a two-electrode system was used in which the titanium foil was used as a working electrode, platinum mesh as a counter electrode. Scanning electron microscopy (FESEM, JEOL JSM-7600F, Tokyo, Japan) equipped with energy-dispersive X-ray spectroscopy (EDS, INCA, Oxford Instruments, Oxford, UK) was used to image the formed microstructures.

### 2.3. Thermal Modification of Titanium Dioxide Nanotubes

TNTs layers were thermally modified in an annealing furnace from AMP company (Zielona Gora, Poland) in an argon atmosphere at 450 °C for 2 h, with the heating and cooling rate of 6 °C·min^−1^ [7]. Annealing allowed the transformation of the amorphous form usually present in TNTs to the crystalline form of rutile and anatase.

### 2.4. Modification of TNTs with Gold Nanoparticles

#### 2.4.1. Physical Adsorption Methods (PA)

##### Direct Adsorption

The AuNPs were immobilized onto titanium dioxide nanotubes electrodes using a simple PA process according to research carried out by Macak et al. [14]. After the deposition of AuNPs, the electrode was taken out and washed thoroughly with distilled water. After this step, the samples were dried under a liquid nitrogen atmosphere.

##### Soaking at High Temperature

The titanium dioxide nanotubes electrode was soaked in the gold nanoparticles solution and heated at atemperature of 140 °C for one hour (standard time of soaking process). After the deposition, the samples were taken out and washed with distilled water and next, the samples was dried under liquid nitrogen atmosphere.

The deposition procedure was carried out in accordance with Elmoula et al. [15].

#### 2.4.2. Electrodeposition Methods

The electrodeposition methods were performed in a three-electrode configuration were titanium dioxide nanotubes were used as a working electrode, platinum mesh as a reference electrode and silver/silver chloride electrode (Metrohm, Herisau, Switzerland), *E*_Ag/AgCl_ = 0.222 V vs. standard hydrogen electrode) as a reference electrode. After deposition, the AuNPs/TNTs were washed using distilled water, and then dried under liquid nitrogen atmosphere.

##### Cyclic Voltammetry Method (CV)

Deposition of gold nanoparticles through CV scans from −1.25 V to −0.7 V (vs. Ag/AgCl) with a scan rate of 0.05 V/s was performed in 0.01 M PBS (3 mL, pH 7.4) containing different concentration of tetrachloroauric acid 0.1 mM, 1 mM, 5 mM, 10 mM. The process was carried out for 40 cycles.

In addition, TNTs with a micro-gold layer (Au/TNTs), were used to compare the electrochemical parameters of the gold layer and gold nanoparticles. The so-called micro-gold layer was produced in the deposition parameters described above, but for 150 cycles.

##### Chronoamperometry Method (CA)

Deposition of AuNPs on TNTs using chronoamperometry method with the potential of −0.9 V was performed in 1 mM solution of HAuCl_4_, for different time of deposition process 60 s, 120 s, 180 s, 240 s and in 10 mM solution of HAuCl_4_ for 60 s of deposition.

### 2.5. Deposition of BSA onto AuNPs/TNTs

BSA was dissolved in 0.01 M PBS. The deposition process included dropping 5 µL of solution BSA with a concentration of 1 mg/mL on the TNTs and TNTs with nanoparticles and AuNPs/TNTs produced using CA and CV methods. The deposition of BSA process was carried outat 40 °C for 1 h according to the procedure described by Kopac et al. [23]. The samples were then rinsed three times using an electromagnetic stirrer in 3 mL PBS for 30 s. The electrochemical characteristics were performed three times, after each rinse step to assess the stability of BSA binding to the electrode substrate. The adsorption of BSA on the TNTs and on the AuNPs/TNTs was rated based on values of relative change (%) in the impedimetric parameters.

The next step was to evaluate the selectivity of the developed platform to BSA-anti-BSA complex by checking the changes of the impedimetric characteristic of AuNPs/TNTs when two antigens coexist, i.e., BSA and IL-6. For this, the protein immobilization was carried out for 1 h at 40 °C in the same concentration (1 mg/mL).

### 2.6. Characteristics of Morphology, Chemical and Phase Composition of TNTs and AuNPs/TNTs

A SEM equipped with energy-dispersive X-ray spectroscopy was used to investigate the microstructure and chemical composition of TNTs and AuNPs/TNTs. X-ray diffraction spectroscopy was performed at a Panalytical Empyrean diffractometer (Malvern Panalytical, Malvern, UK). For research, Cu Kα radiation at 40 kV and 40 mA (λ = 1.540508 Å) was used. The analysis performed with an X-ray photoelectron spectroscope carried out in a PHI Versa ProbeII Scanning XPS system (ULVAC-PHI, Chigasaki, Japan), where the scan area was 400 µm × 400 µm, the pass energy was 46.95 eV. The photoelectron take off angle was 45°.

### 2.7. Electrochemical Measurements

Open circuit potential (OCP) and electrochemical impedance spectroscope (EIS) measurements for annealed TNTs, AuNPs/TNTs before and after the BSA deposition process were conducted using an electrode configuration described in Section 2.4.2. The open-circuit potential was measured at room temperature for 1800 s. Electrochemical impedance investigations were carried out in a frequency from 0.1 to 10^5^ Hz (signal amplitude of 0.01 V). The described electrochemical measurements were carried outin phosphate-buffered solution (0.01 M, 20 mL, pH 7.4).

All of the described measurements were repeated three times. The Bode and Nyquist plots presented in Appendix A show the closest to the calculated from three samples and repeated average value of the curves. The obtained equivalent circuit was selected with Nova 2.1.4. For all of the electrochemical measurement the potentiostat/galvanostat model PGSTAT 302N from Autolab (Metrohm, Herisau, Switzerland) was used.

## 3. Results

### 3.1. Characterization of TNTs before and after Micro Gold Layer Deposition

Scanning electron micrographs (Figure 1a,c) show titanium dioxide nanotubes produced according to the procedure described in Section 2.2. The produced nanotubes had a 50 ± 5 nm—diameter and a height of 1000 ± 10 nm. The titanium dioxide nanotubes were not damaged in the annealing processat 450 °C for 2 h. Figure 1b shows an irregular micro gold layer, which completely closes the nanotubular morphology of TNT arrays.

Annealing of TNTs (Figure 2) caused the changing amorphous form of titanium dioxide nanotubes into the crystalline form [24,25]. Annealing results in a formation of oxygen vacancies, which improve TNTs conductivity and increase the electron transfer as a result of the conversion of Ti^4+^ to Ti^3+^ [9,10]. The prominent diffraction peak located at 25.4° corresponds to anatase, the peak located at 28.0° is attributed to the rutile form of TiO_2_ [26]. The obtained results (Figure 2) confirmed that annealing of titanium dioxide nanotubes resulted in anatase predominance [26]. Gold exhibited four peaks: 2θ = 38.1, 64.5 and 77.7. All the peaks corresponded to standard Bragg reflections (111), (220) and (311) [27].

XPS analysis of the bare titanium dioxide nanotubes and after modification with the micro-gold layer is shown in Figure 3. Surface concentrations of elements obtained from XPS measurements for TNTs and Au/TNTs are presented in Table 1. Carbon presence can be attributed to the adventitious carbon usually found on the surface of most air-exposed samples. The C 1s spectra for all samples can be fitted with three components a rising from aliphatic carbon (284.8 eV), C–O bonds (the line at 286.1 eV) and O–C=O groups evidenced with the line centered at ~289.0 eV [28]. The presence of TiO_2_ and Ti_2_O_3_ in the nanotube layer was confirmed. According to reference [29] the standard of the peak of Ti 2p3/2 in TiO_2_ for Ti^3+^ is located at 457.7 eV, while for Ti^4+^ is at 459.5 eV. The O 1s peak for TiO_2_ is located 529.3 eV [22,29,30]. Analysis of the values presented in Table 1 shows that the smaller amount of oxygen at the surface (at 8.8%) and higher amountof oxygen inside the oxide film (lattice oxygen–at 47.1%) was observed. Annealing of titanium dioxide nanotubes caused the lack of oxygen in the surface. The oxygen vacancies improved the electrical conductivity of TNTs [9,10]. Annealing caused increases the peak value when compared to that associated with the unmodified samples. The shift of O 1s and Ti 2p to lower binding energies was observed as a result of samples thermal treatment. A shift of Ti^3+^ results from the migration of titanium and oxygen ions to electron traps. The shift in O 1s spectra can be attributed to the transfer of electrons to the neighboring oxygen vacancies [30]. For Au/TNTs, the spectra can be fitted with one 4f doublet structure. The main 4f_7/2_ line is centered at 84.0 eV for the sample, which indicates metallic state of gold (Au^0^) in thin-film or bulk [31].

Electrochemical analysis of TNTs after annealing and TNTs with a micro-gold layer will be used as reference samples later in this publication (Section 3.3, Section 3.4 and Section 3.5).

### 3.2. Characterization of TNTs after Gold Nanoparticle Deposition Using Physical Adsorption Methods

SEM micrographs of the TNTs surface after the deposition of gold nanoparticles using physical adsorption methods described in Section 2.4.1 are shown in Figure 4.

Two mechanisms of gold nanoparticles formation and deposition are possible to modification of titanium dioxide nanotubes surface. First, the AuNPs are formed in selected solutions and next deposit to the surface of the TNTs. In the second mechanism, nanoparticles easily nucleate directly on various sites of the surface of nanotubes. The non-covalent interactions between titanium dioxide nanotubes and gold nanoparticles include electrostatic interactions, p-p stacking, hydrophobic interactions or van der Waals interactions. Electrostatic interactions are particularly important due to their long-range and reversible. In our studies, due to the charge of surface, we used electrostatic interactions. According to the author’s previous studies [22], the AuNPs in PBS solution have positive charge value. According to Table 2, the surface charge of TNTs is equal to −47 ± 7.2 mV. Thus, regarding the research presented by Yu et al. [32], it is possible to use electrostatic interactions to deposit gold nanoparticles on the surface of titanium dioxide nanotubes. Positively charged gold nanoparticles are attracted to nanotubes, which have a negative charge.

Studies have shown that no gold nanoparticles are deposited on TNTs as a result of the direct adsorption method (Figure 4a) differently to the studies described in the literature [14]. For the same parameters of the deposition process, Macak et al. [14] obtained well distributed and well adhered on the TiO_2_ surface gold nanoparticles (diameter of 3 ± 2 nm). Functionalization using soaking at a high temperature of 140 °C, according to the research described by Elmoula et al. [15], allows to deposition of AuNPs on TNTs. However, the deposited nanoparticles had a wide range of diameter (5–20 nm), non-spherical shape and precipitated a few hours after the deposition process. To date, the stability of the deposited nanoparticles over time has not been demonstrated in the literature. In connection with the above, we excluded these methods from further research.

### 3.3. Characterization of AuNPs/TNTs after AuNPs Deposition Using Cyclic Voltammetry Method

Figure 5a–d shows the images of TNTs after the deposition of gold nanoparticles using CV method. The deposition process was carried out 40 cycles and different concentrations of HAuCl_4_. The choice of the number of deposition for cycles was justified in the authors’ previous publication [22]. The AuNPs/TNTs for which deposition was carried out for 40 cycles had the positive value of OCP, one of the lowest values of SD (standard deviations), impedance modulus and time constant. This indicates that the best electron transfer was obtained for 40AuNPs/TNTs [22].

The samples presented on Figure 5a–d were determined as 0.1 mM AuNPs/TNTs, 1 mM AuNPs/TNTs, 5 mM AuNPs/TNTs, 10 mM AuNPs/TNTs.

Dissociation of the electrolyte of gold salt solution [AuCl_4_]^−^ was presented by Equation (1). As shown in the Equation (2) and according to the data presented by [21], Au^3+^ can be reduced to Au^0^ and obtain electrons.

[AuCl_4_]^−^ ↔ Au^3+^ + 4Cl^−^(1)

Au^3+^ + 3e ↔ Au^0^(2)


The reduction of Au^3+^ concentration causes the concentration gradient between titanium dioxide nanotubes and the solution. On the other hand, the occurrence of the concentration gradient causes the moves of Au^3+^ ions into TNTs and the formation of many reduced Au^0^ crystals [21]. Higher current densities occurring at the places of TNTs contact cause the deposition of AuNPs in these places [21,22]. This causes uniform distribution of AuNPs on titanium dioxide nanotubes. As the concentration of tetrachloroauric acid increases, the nanoparticles ability to aggregate increases (Figure 5b–d).

Table 2 shows the diameter of AuNPs and OCP average values for titanium dioxide nanotubes, micro-gold layer, 0.1 mM–10 mM AuNPs/TNTs. As it can be seen, the increase of HAuCl_4_ concentration from 0.1 mM to 10 mM caused the increase of the diameter of the deposited gold nanoparticles from 20.3 ± 2.9 nm for 0.1 mM HAuCl_4_ to 182.3 ± 51.7 nm for 10 mM HAuCl_4_. However, increase of the HAuCl_4_ concentration causes increase of SD values for diameter of nanoparticles due to the formation of many agglomerates on the TNTs. After the gold nanoparticles deposition process, the values of the open-circuit potential of the samples increases compared to bare TNTs. It is a result of the inherent inertness of gold and the fact that it is not uniformly distributed on the surface of TNTs [21]. According to the theory of electrostatic binding, the aim is to obtain a positively charged surface of TNTs, which will facilitate the binding of negatively charged proteins [33].

The results of impedance measurements (Nyquist and Bode plots are presented in Appendix A) were fitted to the equivalent circuit shown in Figure 6.

The selected equivalent circuit represents the reactions occurring at the titanium dioxide nanotubes/electrolyte and gold nanoparticles/titanium dioxide nanotubes/electrolyte interface.

The Rs element corresponds to the resistance of the solution. Because of the of TNTs formation on the surface of titanium foil leading to increased heterogeneity, the constant phase element (Q) was introduced [34,35]. The parallel combination (R1Q1) corresponds to the resistance of TNTs and AuNPs/TNTs and the capacitance C1 of these samples was used. Following parameters (R2Q2), in turn, describe the properties of the TiO_2_ porous barrier layer. The samples capacitance values were detonated as a C1, C2 and determined from Equation (3).
(3)$$C=\frac{{\left(RQ\right)}^{1/N}}{R}[F]$$
where N is an empirical constant, ranging from 0–1.

The values determined by fitting the EIS characteristic are presented in Table 3. The samples for which the deposition process was performed in a lower concentration of HAuCl_4_ (0.1 mM, 1 mM, 5 mM) showed a decrease in the value of charge transfer resistance compared to bare TNTs. This proved that the AuNPs promotes the electron transfer. The τ (time constant) calculated from R1C1 increased with the gold salt solution concentration increase. The τ increase was accompanied by increasing the AuNPs agglomerates formation on TNTs, which limited the pathways for the electrons. Moreover, deposition of AuNPs onto TNTs resulted in a decrease of the solution resistance. Importantly, the most significant increases were observed for AuNPs/TNTs without agglomerates, especially for 0.1 mM AuNPs/TNTs. The N2 value is near 1, which confirmed that the obtained surface is smooth. The lower value also proved the deviation from ideal capacitive behavior [36]. The N2 value close to onehad TiO_2_ nanotubes before modification with AuNPs and with the least amount of agglomerates the deposition of gold nanoparticles was performed in 0.1 mM and 1 mM tetrachloroauric acid. The production of the micro-gold layer on the TNTs increased the resistance Rs of the surface and caused a decrease in the capacity of the layer of titanium dioxide nanotubes due to the reduction of the surface.

According to Table 2, the TNTs have positive OCP values after the deposition of gold nanoparticles. This is especially important in the case of adsorption of biologic elements such as bovine serum albumin, which has a negative electrical charge (isoelectric point of BSA-4.7) in solution of pH 7.4 [37]. Among these electrodes, the lowest value of τ, indicating the most effortless electron transfer between the electrode and the electrolyte, is characterized by the surface after deposition AuNPs for 40 cycles in 0.1 mM HAuCl_4_ (Table 3).

The use of gold nanoparticles for the deposition of well-spherical geometry of a 0.1 mM solution avoids stabilizers. The micro-gold layer was excluded from further analysis because it caused a reduction in the surface area of titanium dioxide nanotubes.

### 3.4. Characterization of AuNPs/TNTs after Gold Nanoparticles Deposition Using Chronoamperometry Method

Figure 7a–d shows the surface morphologic characteristics of the AuNPs on the TNTs surface observed through SEM. AuNPs were deposited on the TNTs at −0.9 V with a different time of process duration in 1 mM HAuCl_4_ solution. The samples were marked accordingly as xAuNPs/TNTs, x is the time of the deposition process (60 s, 120 s, 180 s, 240 s).

AuNPs are evenly distributed on the surface of the TNTs. The AuNPs nucleate at the contact boundaries of nanotubes.

Table 4 shows the height of TNTs, the diameter of AuNPs and open-circuit potential average values. As it can be seen, as the gold deposition time increases, the titanium dioxide nanotube layer dissolves. The process of chronoamperometric AuNPs deposition therefore causes destruction of the TNTs layer due to the electrolyte solution’s interaction. This may be attributed to the chemical dissolution of TiO_2_. The weakening of the Ti–O bond may result in the collapse of the TNTs structure [38]. Increase the AuNPs deposition time leads to an increase in the amount of nanoparticles (Figure 7). The size of AuNPs does not evidently change (Table 4), according to [21]. This finding indicates that AuNPs does not agglomerate. The gold nanoparticles deposited on the surface of TNTs have a diameter of approximately 35.0 ± 5.0 nm.

The deposition of nanoparticles by chronoamperometry causes an increase in OCP value compared to TNTs from −47 ± 7.2 mV to −7.5 ± 1.1 mV. However, no positive potential values of surfaces were obtained, which is particularly important in protein deposition.

The electrical values obtained by fitting equivalent circuits based on electrochemical parameters presented in Appendix A are shown in Table 5. Improvement of the electrochemical parameters (Rs, R1, C1, R2, C2) was obtained for the titanium dioxide nanotubes after deposition of AuNPs by the chronoamperometry method (deposition time: 60 s). For other deposition times (120 s, 180 s, 240 s) compared to TNTs before modification, the increase in electrolyte resistance, increase in resistance (R1) and a decrease in the capacity of the AuNPs/TNT layers were observed. As the deposition time increased, the time constant decreased from 14.62 s to 4.29 s. This was the consequence of the increase of the high content of gold (Table 4) resulting from the deposition time increase. The 60 s AuNPs/TNTs was selected for further analysis due to the dissolution of TNTs layer.

The next stage of the study was comparing the selected time of CA deposition of gold nanoparticles (60 s) and different concentration of tetrachloroauric acid (1 mM, 10 mM). Increasing the concentrations of HAuCl_4_ causes an increase in the diameter of AuNPs from 36.7 ± 5.9 nm to 76.0 ± 3.0 nm and affects the dissolution oftitanium dioxide nanotubes, which are shortened to 763 ± 75.1 nm (Table 6).

Electrochemical parameters (Table 7) obtained by circuits shown in Figure 7, confirm that the dissolution of the titanium dioxide nanotube layer due to higher concentration of tetrachloroauric acid causes the increase in the resistance of the TNTs layer (6.1 × 10^3^Ω) and the decrease in its electrical capacity (0.61 × 10^−4^ F) compared to the deposition carried out in a solution with a lower concentration, respectively: 4.7 × 10^3^ Ω, 3.11 × 10^−4^ F. This causes the deterioration of the surface electrical conductivity.

### 3.5. Comparison of the Impact of CV and CA Methods of AuNPs Deposition on the TNTs Characteristics and Protein Adsorption

A comparison of the effects of the CA and CV method on the properties of TNTs was carried out for surfaces modified with gold nanoparticles for selected in Section 3.3 and Section 3.4 parameters of the deposition processes: CA-40 cycles of deposition in 0.1 mM HAuCl_4_, 60s of deposition carried out and 1 mM HAuCl_4_.

XRD analyses of the titanium dioxide nanotubes thermally before and after the deposition of AuNPs using CV and CA methods are shown in Figure 8. Similar to the study shown in Figure 2, the presence of anatase and rutile phases of TiO_2_ was confirmed. The obtained results confirm the predominance of anatase in TNTs structure, which has better properties for adsorption of biologic elements, due to higher hydrophilicity [26]. The lower intensity of Ti, anatase and rutile peaks for electrodes after the deposition of gold nanoparticles by chronoamperometry may be the result of the dissolution of the TNTs layer. Higher peak intensities of Au are also observed, especially peaks corresponding to (311) Bragg reflections for deposition carried out by the cyclic voltammetry method.

The results of the XPS analysis of the bare TNTs and TNTs with AuNPs deposited using CA (60 s, 1 mM HAuCl_4_) and CV (40 cycles, 0.1 mM HAuCl_4_) methods are shown in Figure 9. The presence of TiO_2_ and Ti_2_O_3_ in the nanotube layer has been confirmed. According to Table 8, annealing of titanium dioxide nanotubes causes improved its electrical conductivity [25]. According to [39], the shifted 4f7/2 line to lower binding energy indicates the production of nanoparticles with spherical geometry. This is due to the lower binding energy compared to large nanoparticles [39]. XPS analysis showed a higher gold content for samples after deposition by cyclic voltammetry, which also confirms the higher intensity of the gold peaks obtained in the XRD analysis (Figure 8).

The presence of TiO_2_ and Ti_2_O_3_ in the nanotube layer has been confirmed. According to [29] the standard of the peak of Ti 2p3/2 in TiO_2_ for Ti^3+^ is located at 457.7 eV, while forTi^4+^ is at 459.5 eV. The O 1s peak for TiO_2_ is located 529.3 eV [22,29,30]. Analysis of the values presented in Table 1 the higher amount of oxygen inside of the oxide film (lattice oxygen—at 47.1%) was observed. Annealing of titanium dioxide nanotubes caused the lack of oxygen in the surface. The oxygen vacancies improved the electrical conductivity of TNTs [9,10]. The annealing caused increases in the peak value when compared to that associated with the unmodified samples. The shift of O 1s and Ti 2p to lower binding energies was observed as a result of samples’ thermal treatment. A shift of Ti^3+^ is a result of the migration of titanium and oxygen ions to electron traps.

The immobilization of the bovine serum albumin procedure was in accordance with Kopac et al. [23]. In work, the researchers showed that the highest value of the adsorption efficiency (72%) was obtained for the temperature of 40 °C.

Figure 10 shows the results of scanning electron microscopy analysis for titanium dioxide nanotubes before (TNTs) and after the gold nanoparticle modification process (60 s AuNPs/TNTs and 40 AuNPs/TNTs). Gold nanoparticles increase the stationary potential to the positive values and favor the adsorption of proteins. The deposition of BSA proceeds according to three steps: transport protein, deposition and spreading on the surface. The pH value of PBS solution (7.4) affects on the deposition of protein, its value should be 7–8 [40]. The bovine serum albumin due to it isoelectric point (4.7) has a negative charge in phosphate buffered saline. The increase in BSA adsorption is the result of electrostatic interactions described in Section 3.2 of this manuscript. The electrostatic interaction hypothesis confirms the strong bond between negatively charged BSA and positively charged gold nanoparticles. In this hypothesis, the protein attaches itself to the passivating layer on the gold surface, with little direct interaction between the BSA and the gold surface [40].

Figure 11 presents a diagram of percentage change of the impedance parameters for TNTs, 40 AuNPs/TNTs and 60 s AuNPs/TNTs after immobilization of BSA for one hour.

The calculated percentchanges of electrochemical values are presented in Figure 11. For TNTs, 40 AuNPs/TNTs and 60 s AuNPs/TNTs the increase in electrochemical parameters (impedance modulus, imaginary part of impedance and real part of impedance) as a results of BSA deposition was observed. The highest increases of values were noted for TNTs after modification with AuNPs, especially for those produced by the CV method. The obtained results confirm that AuNPs promote protein adsorption [40]. Using CV, we obtained almost a 5.5-fold (324%) increase in ReZ, compared to TNTs (59%). High values of standard deviations calculated for platform modification with AuNPs, may indicate that such large molecules like BSA do not cover AuNPs/TNTs uniformly, so the number of molecules adsorbed on them is very random. Hence, the measured parameters on the same samples may differ from each other [40,41]. For this reason, nanoparticles with larger diameter (36.7 ± 5.9 nm) produced by chronoamperometry have a higher standard deviation. Based on the change in impedance parameters after washing the samples three times, it was found that the Au nanoparticles promoted the stability of BSA binding to the electrode substrate.

Interleukin-6 was used as a comparative protein in order to evaluate the selectivity of the bare titanium dioxide nanotubes and TNTs after modification with gold nanoparticles using chronoamperometry and cyclic voltammetry methods to BSA-anti-BSA complex. As shown in Figure 12, larger changes in impedance parameters were noted for the 40 AuNPs/TNTs, so it is the best platform for biosensing applications. It is likely to be more sensitive than the unmodified TNTs and TNTs modified using the chronoamperometry method. For this method, the ReZ parameter increased almost twice than the unmodified TNTs. This allows us to conclude that higher gold content—and also smaller diameter particles—favor BSA adsorption. The change of impedance parameters on AuNPs/TNT electrodes toward BSA were the highest—about 2-fold higher than for TNTs and 60 s AuNPs/TNTs, and approximately 3-fold higher for 40 AuNPs/TNTs than anti-BSA. Changes in these parameters in the case of immobilization of IL-6 are smaller than the standard deviation, so it can be assumed that there was no nonspecific binding of IL-6 to the TNTs and AuNPs/TNTs platform.

## 4. Conclusions

The study aimed to compare the microscopic, chemical, phase and electrochemical characteristics of bare titanium dioxide nanotubes and TNTs modification with AuNPs deposited using cyclic voltammetry and chronoamperometry methods. To do so, several gold nanoparticles deposition methods were selected: physical adsorption, cyclic voltammetry and chronoamperometry. Then, the most critical process parameters for the conductivity of TNTs were selected. XRD analysis results confirmed a higher content of anatase phases than rutile due to annealing of TNTs carried out at 450 °C. We found that, contrary to literature reports [14,15], direct adsorption methods should not be used to deposition gold nanoparticles on TNTs substrate. It is important that AuNPs deposited by soaking at high temperature methods are not stable and AuNPs precipitate shortly after their production. The research found that using high dispersion and spherical on AuNPs causes improved the capacitance of TNTs. The AuNPs increase the corrosion resistance of TNTs [19]. The use of AuNPs produced by the CV method showed that as with an increase of gold salt solution concentration, the diameter of the gold nanoparticles and the number of agglomerates increases. In turn, deposition carried out by chronoamperometry causes an increase in the amount of gold, but the diameter of the nanoparticles does not increase. An increase in the concentration of HAuCl_4_ from 0.1 mM to 10 mM causes an increase in the AuNPs diameter. As a result of depositing NPs using the chronoamperometry method, the nanotubes layer also dissolves. It was shown that with an increase in the duration of the process—as well as the concentration of tetrachloroauric acid—the layer of titanium dioxide nanotubes dissolves, which so far has not been studied in the literature. For the selected process parameters CV (40 deposition cycles 0.1 mM HAuCl_4_) and CA (60 s, 1 mM HAuCl_4_), it was found on the basis of XPS analysis that both of these methods allow producing spherical nanoparticles. Nanoparticles promote the adsorption of biologic elements. The highest increase in electrochemical parameters resulting fromthe BSA deposition process was observed for TNTs modification with AuNPs using the CV method. The present work confirms that AuNPs promote the adsorption of protein [40]. The CV method obtained almost a 5.5-fold (324%) increase in ReZ, compared to TNTs (59%).

This study confirmed the possibility of using the AuNPs to improve the sensitivity of the TNTs and allows to use this platform in label-free detection system.

## Figures and Tables

**Figure 1 materials-13-04269-f001:**
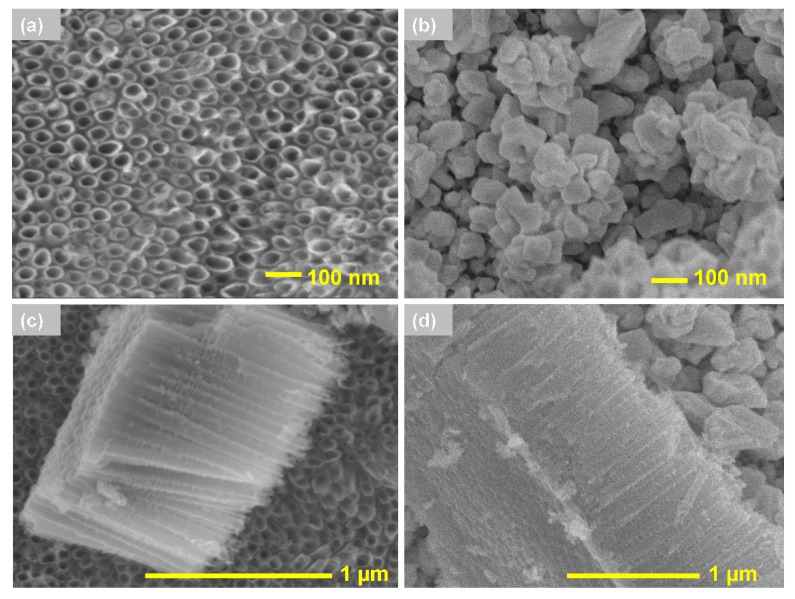
Images of annealed titanium dioxide nanotubes (**a**) before and (**b**) after micro-gold layer (Au/TNTs) deposition with a (**c**) cross-section of TNTs and (**d**) TNTs with the micro-gold layer.

**Figure 2 materials-13-04269-f002:**
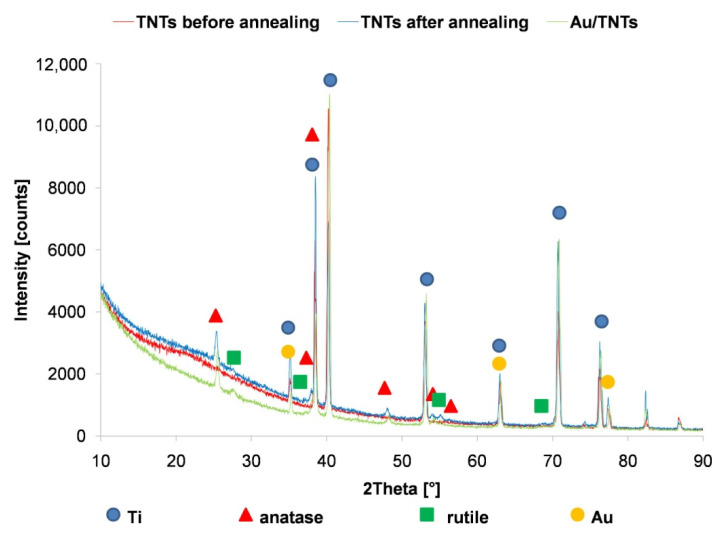
XRD patterns of titanium dioxide nanotubes before and after thermal modification and micro-gold layer (Au/TNTs) deposition.

**Figure 3 materials-13-04269-f003:**
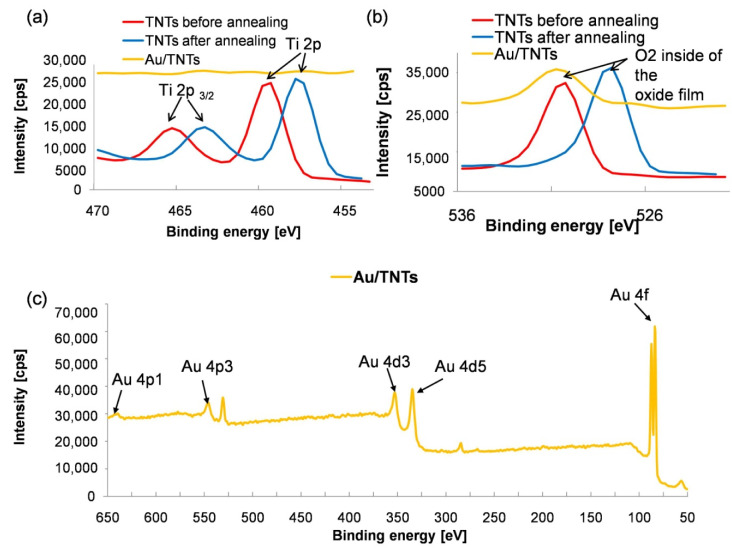
Chemical composition of unmodified, annealing and gold-doped TNTs measured with XPS (**a**) Ti 2p (**b**) O 1s and (**c**) Au spectra.

**Figure 4 materials-13-04269-f004:**
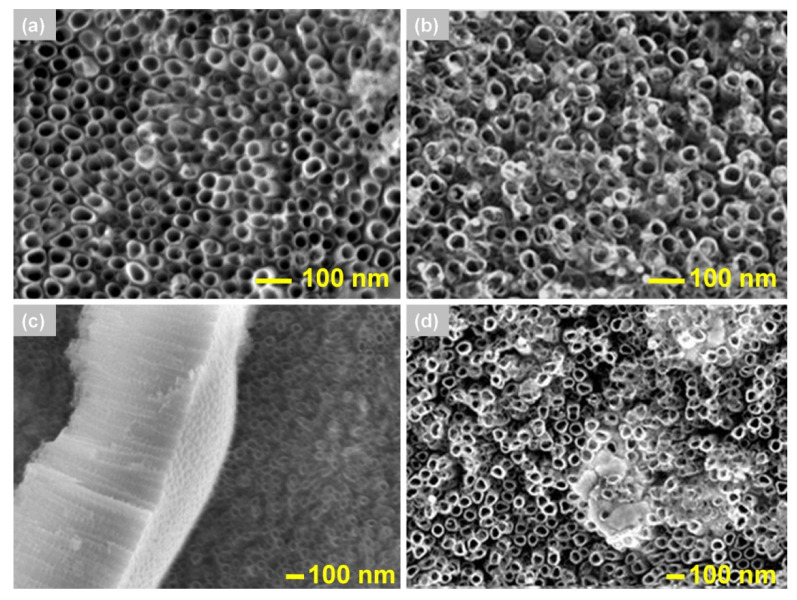
Micrographs of TNTs after AuNPs deposition process using different methods. (**a**) Direct adsorption with (**c**) cross-section of TNTs, (**b**) soaking at high temperature—10 min after deposition, (**d**) 3 h after deposition.

**Figure 5 materials-13-04269-f005:**
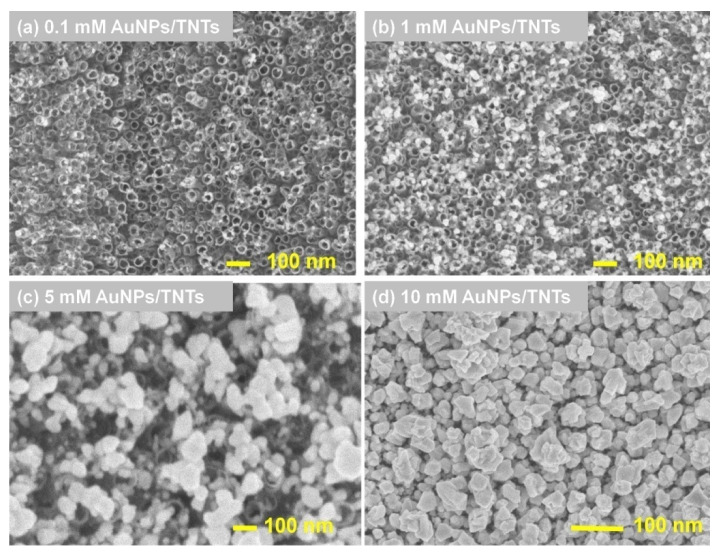
Micrographs of titanium dioxide nanotubes after AuNPs deposition using cyclic voltammetry method—40 cycles in (**a**) 0.1 mM, (**b**) 1 mM, (**c**) 5 mM, (**d**) 10 mM HAuCl_4_.

**Figure 6 materials-13-04269-f006:**
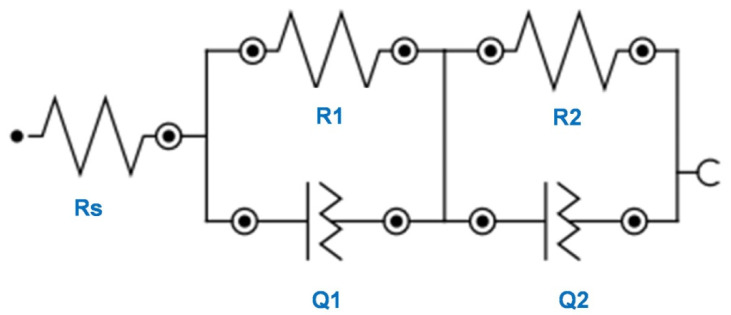
Equivalent circuit used to model the impedance spectra of the TNTs and AuNPs/TNTs.

**Figure 7 materials-13-04269-f007:**
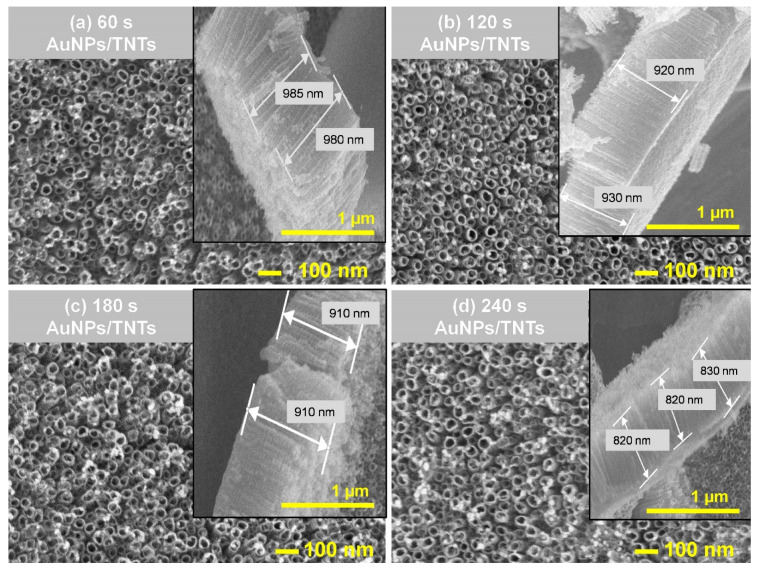
SEM images of TNT arrays after gold nanoparticles with cross-section (insets) deposition using CA method carried out in 1 mM HAuCl_4_ for (**a**) 60 s, (**b**) 120 s, (**c**) 180 s, (**d**) 240 s.

**Figure 8 materials-13-04269-f008:**
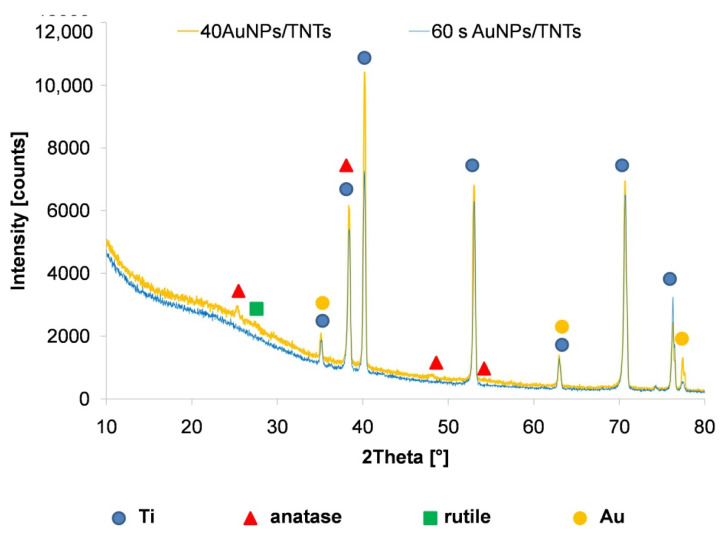
XRD patterns of annealed TNTs before and after AuNPs deposition using cyclic voltammetry and CA methods.

**Figure 9 materials-13-04269-f009:**
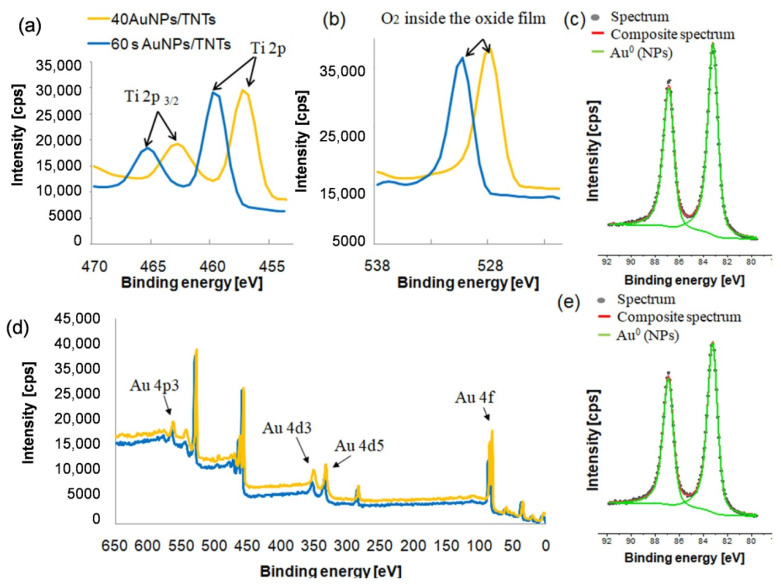
Chemical composition of TNTs surfaces before and after AuNPs deposition using CV (40 AuNPs/TNTs) and CA (60 s AuNPs/TNTs) methods (**a**) Ti 2p, (**b**) O 1s (**c**–**e**) Au spectra.

**Figure 10 materials-13-04269-f010:**
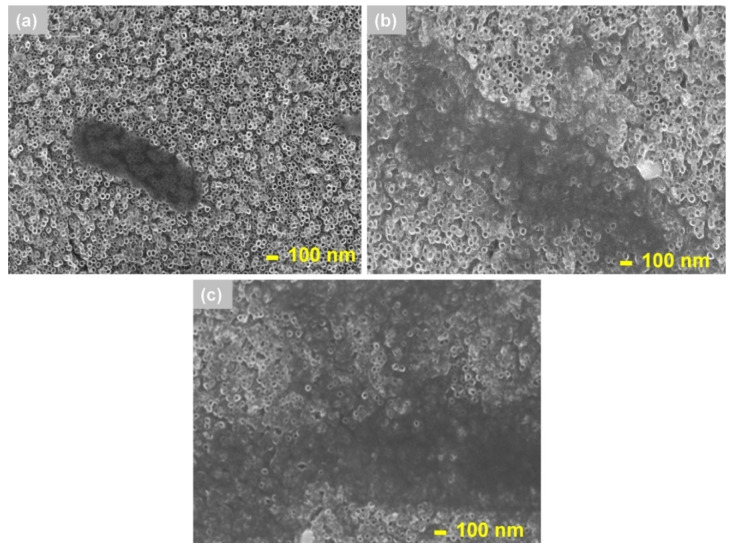
SEM images of the (**a**) TNTs, (**b**) 60 s AuNPs/TNTs and (**c**) 40 AuNPs/TNTs after bovine serum albumin deposition carried out for 1 h.

**Figure 11 materials-13-04269-f011:**
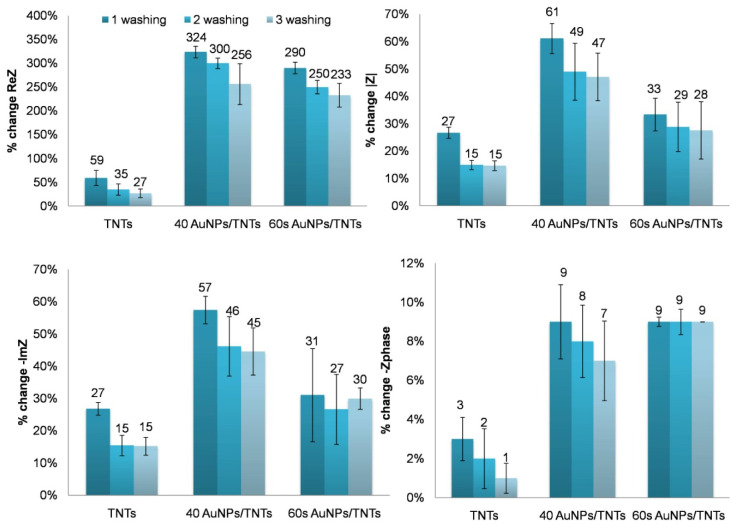
Values of percent change of electrochemical parameters due to one hour of bovine serum albumin deposition on the TNT arrays, 40 AuNPs/TNTs and 60 s AuNPs/TNTs.

**Figure 12 materials-13-04269-f012:**
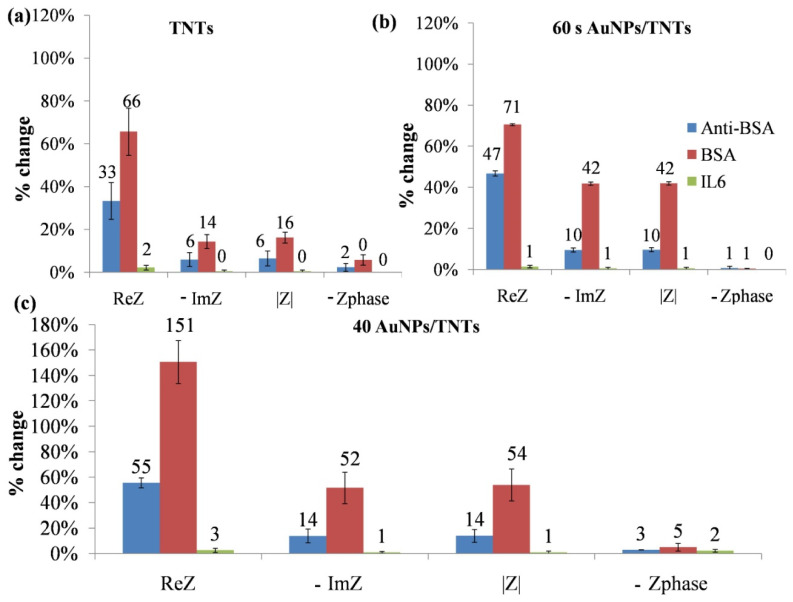
Percentage change of the impedance parameters resulting from anti-BSA, BSA and IL-6 immobilization on (**a**) TNTs, (**b**) 60 s AuNPs/TNTs and (**c**) 40 AuNPs/TNTs.

**Table 1 materials-13-04269-t001:** Composition (atomic%) determined by fitting XPS data for TNTs and Au/TNTs.

TNT Arrays	Element	C			O		Ti	Au
	Compound	C–C	C–O	O–C=O	MeOx (Lattice)	MeOx (Defective)	TiO_2_	Au^0^
Before annealing	TNTs	15.8	4.9	1.2	39.2	10.6	21.3	0.0
After annealing	TNTs	16.8	4.2	1.5	47.1	8.8	21.7	0.0
	Au/TNTs	22.4	6.0	5.0	12.0	20.5	0.0	28.5

**Table 2 materials-13-04269-t002:** OCP values and diameter of AuNPs determined for titanium dioxide nanotubes and titanium dioxide nanotubes with a micro-gold layer and AuNPs. Values indicate means ± SD.

Samples	TNTs	0.1 mM AuNPs /TNTs	1 mM AuNPs /TNTs	5 mM AuNPs /TNTs	10 mM AuNPs /TNTs	Au/TNTs
AuNPs diameter (nm)	-	20.3 ± 2.9	26.6 ± 4.8	99.4 ± 42.4	182.3 ± 51.7	-
OCP (mV) vs. Ag/AgCl	−47 ± 7.2	25 ± 8.6	51 ± 2.6	93 ± 1.6	110 ± 10.8	148 ± 23.1

**Table 3 materials-13-04269-t003:** Values obtained by fitting an equivalent circuit for TNTs and AuNPs/TNTs.

Electrical Parameters	TNTs	0.1 mM AuNPs/ TNTs	1 mM AuNPs/ TNTs	5 mM AuNPs/ TNTs	10 mM AuNPs/ TNTs	Au/TNTs
Rs (Ω)	25.1	8.59	13.45	20.10	24.01	38.2
R1 × 10^3^ (Ω)	6.15	4.55	7.12	16.15	21.11	4.81
C1 × 10^−3^ (F)	9.61	2.66	2.08	4.12	6.52	3.55
N1	0.97	0.98	0.98	0.96	0.96	0.91
τ = R1C1 (s)	49.01	12.10	14.81	66.54	137.64	17.08
R2 × 10^3^ (Ω)	42.10	33.70	39.91	53.01	56.01	37.02
C2 × 10^−4^ (F)	2.11	4.72	4.14	3.21	3.25	1.51
N2	0.95	0.98	0.98	0.96	0.93	0.81
Χ^2^	0.0073	0.0069	0.0065	0.0044	0.0075	0.0095

**Table 4 materials-13-04269-t004:** Average height of TNTs, AuNPs diameter and OCP for TNTs and AuNPs/TNTs. Values indicate means ± SD.

Sample	TNTs	60 s AuNPs/TNTs	120 s AuNPs/TNTs	180 s AuNPs/TNTs	240 s AuNPs/TNTs
Height of TNTs (nm)	1000 ± 10.0	980 ± 23.1	920 ± 7.1	907 ± 5.8	823 ± 15.3
AuNPs diameter (nm)	–	36.7 ± 5.9	32.9 ± 5.5	33.6 ± 3.0	36.7 ± 5.6
Au (wt%)	–	1.39 ± 0.19	1.48 ± 0.20	1.69 ± 0.14	2.33 ± 0.30
OCP (mV) vs. Ag/AgCl	−47 ± 7.2	−38.7 ± 3.2	−17.0 ± 2.6	−7.5 ± 1.1	−11.1 ± 6.6

**Table 5 materials-13-04269-t005:** Value of equivalent circuit elements for titanium dioxide nanotubes before and after AuNPs deposition using chronoamperometry (CA) method.

Electrical Parameters	TNTs	60 s AuNPs/TNTs	120 s AuNPs/TNTs	180 s AuNPs/TNTs	240 s AuNPs/TNTs
Rs (Ω)	25.1	22.8	31.2	33.2	33.6
R1 × 10^3^ (Ω)	5.2	4.7	5.4	5.6	5.8
C1 × 10^−3^ (F)	9.61	3.11	0.89	0.85	0.74
N1	0.97	0.97	0.96	0.96	0.96
τ = R1C1 (s)	49.01	14.62	4.81	4.76	4.29
R2 × 10^3^ (Ω)	42.1	37.9	55.1	59.5	61.2
C2 × 10^−4^ (F)	2.11	4.30	3.12	2.92	2.88
N2	0.95	0.96	0.95	0.95	0.95
Χ^2^	0.0073	0.099	0.085	0.084	0.082

**Table 6 materials-13-04269-t006:** Average height of TNTs, AuNPs diameter and OCP for TNTs after AuNPs deposition using chronoamperometry method for different concentrations of HAuCl_4_. Values indicate means ± SD.

Samples	Height of TNTs (nm)	Diameter of AuNPs (nm)	OCP (mV) vs. Ag/AgCl
60 s AuNPs/TNTs 1 mM HAuCl_4_	980 ± 23.1	36.7 ± 5.9	−38.7 ± 3.2
60 s AuNPs/TNTs 10 mM HAuCl_4_	763 ± 75.1	76.0 ± 3.0	63 ± 12.8

**Table 7 materials-13-04269-t007:** Value of circuit equivalent elements for TNTs before and after AuNPs deposition using CA method for different gold salt solution concentrations.

Samples	Rs (Ω)	R1 (Ω)	C1 (F)	N1	τ = R1C1 (s)	R2 (Ω)	C2 (F)	N2	Χ^2^
60 s AuNPs/TNTs 1 mM HAuCl_4_	22.8	4.7 × 10^3^	3.11 × 10^−3^	0.97	14.62	37.9 × 10^3^	4.30 × 10^−4^	0.96	0.099
60 s AuNPs/TNTs 10 mM HAuCl_4_	38.1	6.1 × 10^3^	0.61 × 10^−3^	0.96	3.71	62.33 × 10^3^	2.71 × 10^−4^	0.95	0.089

**Table 8 materials-13-04269-t008:** Surface composition (atomic%) determined by fitting XPS data for TNTs before and after AuNPs deposition using CA method.

Element	C			O		Ti	Au
Compound	C–C	C–O	O–C=O	MeOx (Lattice)	MeOx (Defective)	TiO_2_	Au
40 AuNPs/TNTs	19.1	0.8	1.4	43.6	8.0	21.0	6.1
60 s AuNPs/TNTs	18.9	1.4	1.6	43.4	8.8	21.4	4.5

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
