# Peer review of "Comparison of Gold Nanoparticles Deposition Methods and Their Influence on Electrochemical and Adsorption Properties of Titanium Dioxide Nanotubes"

_materials, 2020, doi:10.3390/ma13194269_

Round 1
Reviewer 1 Report
The authors presented a thorough study of the modification of TNTs using 
AuNPs that increased electron transfer and corrosion resistance as well as promoted the 
protein adsorption. The TNTs with gold 
nanoparticles can be used to develop an improved biosensing platform.
Although the manuscript is complete in terms of the background, methods and the results, it lacks clarity in many places. I have the following comments on the content:
The authors used the same section number for section 2.4 and 2.5. You can locate them at Line 131 “2.4. Modification of TNTs with gold nanoparticles” and Line141 “2.4. Deposition of bovine serum albumin onto AuNPs/TNTs 
”
Lines 255 and 256 lack clarity: “Studies have shown that no gold nanoparticles have deposited on TNTs as a result of the direct adsorption method (Figure 4a) differently to the studies described in the literature [14].”
Figure 4 (b): For a clearer comparison, I would use the same scale for the inset (100nm) as the comparison is being made across different periods after deposition.
Line 204: Do the authors mean: No damage was observed on the TNT layers after annealing at 
450°C for 2 hours was observed
Line 233: Do the authors mean: lattice oxygen – at 47.1%) and defective oxygen/surface – at 8.8% 
(Line 234)
Line 236: Do the authors mean “Thermal modification increases the peak value when compared to that associated with the unmodified samples “?
The article is also full of grammatical mistakes and improper use of English. Herein I provide just a sample but you can find more mistakes throughout the article:
- The use of “its influence” may need to be replaced with “their influence” in the title? ” Comparison of gold nanoparticles deposition methods and its influence on electrochemical and adsorption properties of titanium dioxide nanotubes “
Line 20: to obtained
Line 40: will directly impacts 

Line 45: the electrical conductive
Line 48: TiO2nanotubes 

Line 382: Thesize
Line 426: tetrachloroauricacid
Line 442: predominanceof, foradsorption
Line 508: voltammetrymethod 

Line 545: dissolveswhich 

Line 479: improper use of English “go pass through” 

Line 511: grammatical mistake “more sensitive comparison to “
Line 205: Figure 1b shows
Line 272: Define SD
Line 328: Define N after equation 3
Line 340: Define Rs as surface resistance
Line 402: Figure 9 is identical to Figure 6 (It shows Nyquist and Bode plots across gold/salt concentrations). According to the figure caption, it should show the plots across different durations not concentrations. Please revise.
Line 407: Grammatical error: “The equivalent circuit are shown in Figure 7”
Line 407: run on sentence “For other deposition times an increase in electrolyte resistance (Rs), increase in resistance (R1) and a decrease in the capacity of the AuNPs/TNT layers compared to TNTs before modification was observed.”
I recommend the manuscript for publication after proofreading it and improve the presentation.
Reviewer 2 Report
This manuscript compared the performance of TNT before and after AuNP modification. Lots of studies have been studied. It is suggested to be published after considering the below items:
- the manuscript is too long. suggest to combine some section, and leave some section in the supporting information.
- in addition, authors finally use the materials to adsorb BSA, which is the purpose from the aspect of application. Why authors use IL-6 as the control analyte since IL-6 is a cytokine which is very expensive.
- It is suggest to develop a biosensor using the prepared materials, and present the performance of the biosensor.
Reviewer 3 Report
It is a good work to achieve precise control of attachment and shape of AuNPs. However, the paper lacks innovation, because many similar works have been published. Some aspects should be taken into account.
- The manuscript needs editing by someone with expertise in technical English editing paying particular attention to English grammar, spelling, and sentence structure so that the goals and results of the study are clear to the reader.
- Will gold be immersed in the inner wall of nanotubes?
- What is the specific surface area of TNT and whether the deposition particles affect the specific surface area.
- What is the motive force of physical adsorption. Adsorption mechanism should be discussed in more detail.
- The title is Comparison of gold nanoparticles deposition methods and its influence on electrochemical and adsorption properties of titanium dioxide nanotubes. There is no relevant content about the effect of deposition particles on adsorption performance.
Reviewer 4 Report
The authors compare various gold nanoparticles deposition methods and its influence on electrochemical and adsorption properties of titanium dioxide nanotubes in this manuscript. The manuscript is well written although some unscientific language and grammatical errors are made. I would recommend acceptance once the following concerns are addressed:
1.) The manuscript is replete with grammatical errors comprising of incorrect use of tense, plurals and verb usage. For example, Page 1, line 40; "...It is expected that increasing the surface area of the support will directly impacts the 40 effectiveness of biochemical reactions." Here instead of impacts it should just be impact. Please correct all these typos or grammatical errors prior to acceptance.
2.) In Fig. 1a, the inset and the main figure are a bit confusing. Usually, the inset figure is at a higher magnification than the main figure. Here it is the other way around. May be having it in a more traditional manner will help in identifying the surface morphology features in the bulk as well as at a magnified scale. Similar is the case with Figure 4.
Round 2
Reviewer 2 Report
Authors have addressed the comments. It is suggested to be published as it is.